



# Lessons learnt from rockfall time series analysis: data collection, statistical analysis and applications

Sandra Melzner[1,2], Marco Conedera[3], Johannes Hübl[2,4], Mauro Rossi[5]

[1]GEOCHANGE Consulting e.U., Klagenfurt/Vienna, 9020/1170, Austria

[2] Institute of Mountain Risk Engineering (IAN), University of Natural Resources and Life Sciences (BOKU), Vienna, 1190, Austria

[3] Research Unit Community Ecology, Swiss Federal Research Institute (WSL), Cadenazzo, 6593, Switzerland

[4] Department of Civil Engineering and Natural Hazards, University of Natural Resources and Life Sciences (BOKU), Vienna, 1190, Austria

[5] Istituto di Ricerca per la Protezione Idrogeologica, Consiglio Nazionale delle Ricerche, Perugia, 06128, Italy

*Correspondence to*: Sandra Melzner (office@geochange-consulting.com)

**Abstract.**

Historical rockfall catalogues are important data sources for the investigation of the temporal occurrence of rockfalls, which

is a crucial information for rockfall hazard and risk assessments. However, such catalogues are rare and often incomplete. Here, we analyse seven catalogues of historical rockfalls in Austria, Italy, and the USA to highlight existing relationships between the data collection and mapping methods and the representativeness of the resulting rockfall records. Heuristic and simple statistically-based frequency analysis methods are applied to describe and compare the different historical rockfall catalogues. Our results show that different mapping strategies may affect the frequency of the assessed rockfall occurrence

and the completeness/representativeness of the related time series of historical rockfalls. We conclude presenting the advantages and limitations of the application of different frequency-based methods for analysing rockfall catalogues and providing recommendations for rockfall mapping. We furthermore present non-parametric statistical methods for dealing with typically small rockfall datasets which are particularly suited for the characterization of basic rockfall catalogues. Such recommendations should help for the definition of standards for collecting and using temporal rockfall data in hazard and risk

assessments.

## 1 Introduction

The investigation of the temporal occurrence of processes is an essential part in a hazard and risk assessment. Temporal analyses may focus on the estimation of rockfall frequency over time, on the analysis of dispositional and triggering factors of rockfalls, or on the medium to long term analysis of rockfall occurrence trends over time. Such statistical frequency analysis

of rockfall catalogues and of rockfall time series provides fundamental information for hazard and risk studies.





However, published statistical temporal analyses of rockfalls (e.g. power law approaches) are usually rare (De Biagi et al., 2017; Melzner et al., 2020) and aim at estimating the annual, seasonal or daily frequency of rockfalls (and the related return period) and at establishing magnitude-frequency relationships (Agliardi et al., 2009; Corominas et al., 2017; Crosta et al., 2015; De Biagi et al., 2017; Moya et al., 2010). Further, only a few landslides or rockfall studies include an estimation of the

quality of the presented data (Guzzetti, et al. 2012, Melzner et al., 2022).

The objective of this study is to stimulate comprehensive rockfall data collections using standardized procedures. For this purpose, we statistically analyse seven catalogues with temporal information on rockfalls collected with different methods in Italy, USA and Austria and discuss the influence of the applied methods on the quality, representativeness, and completeness of the obtained temporal distribution of rockfalls. Advantages and limitations of the different approaches are presented and

discussed, promoting standards in rockfall data collection and providing guidance for the proper usage of the statistical analysis of rockfall time series and for the evaluation of rockfall data samples representativeness.

## 2 Theoretical framework

### 2.1 Definitions

Information about rockfalls is usually stored in form of *inventories*, *catalogues*, and *record*s. According to Melzner et al.

(2020), a *rockfall inventory* is a type of landslide inventory (Guzzetti et al., 2012) that contains geographical and typological information on the source, transport and deposition area of rockfalls. Information about an individual rockfall or multiple rockfalls can be extracted from the rockfall inventory database in form of different thematic *rockfall catalogues*. A *historical rockfall catalogue* lists the time of occurrence of rockfalls, whereas a *historical catalogue of rockfall consequences* is a list of information on the damage, fatalities, and casualties of rockfall events over time. A *historical rockfall record* or a *rockfall time*

*series* show rockfalls over time for a given region, detailed spatial information is not always available. Rockfall time series, are normally discontinuous with their records separated by irregular time spacings.

The temporal occurrence of rockfall is a fundamental information to assess the *rockfall risk*, which is the probability that a specific hazard will cause harm. Specifically, rockfall risk can be achieved by investigating the frequency and consequences of the rockfall events. The *consequences of a rockfall* can be expressed by the severity of damage on structures and

infrastructures and the impact on population in terms of number of fatalities and/or causalities.

According to Melzner et al. (2020), possible criteria for the evaluation of the *quality* of a rockfall catalogue are the *amount and level of detail of the data* and the *thematic variability of the information*. The level of *comprehensiveness of the catalogue* can be described with *completeness* and the *representativeness* of the collected data and the variability of the geographical location of the single of multiple features encompassed by a rockfall: *Completeness* refers to the proportion of rockfalls

contained in the catalogue with respect to the total number of occurred rockfalls. *Representativeness* of a given rock fall sample/subset refers to the degree with respect to the entire rockfall catalogue from which it is derived i.e., unbiased statistical





inference of the whole population. *Thematic variability* refers to the amount of imprecision of the identification and classification of a rockfall or a given rockfall feature. *Geographic variability* refers to the amount of imprecision of the graphical representation of a rockfall feature to the real geographic position in the study area.

Rockfall time series can be analysed by identifying change points or structural breaks. *Change points* mark abrupt or step changes (i.e., a shift to a higher or lower value) in the mean frequency of documented rockfalls (Pettitt, 1979; Xie et al., 2013). Instead, *structural breaks* are changes over time in the parameters of a regression model fitting the values of a rockfall time series.

## 2.2 Data collection

The frequency of occurrence of rockfalls can be determined (i) qualitatively in the field (Melzner, 2015; Melzner et al., 2019; Stock et al., 2011; Wieczorek et al., 1992), (ii) quantitatively based on historical data on rockfalls or monitoring data (e.g., Janeras Casanova et al., 2017; Katz et al., 2011; Luckman, 2008; Mavrouli and Corominas, 2017; Melzner, 2016; Melzner et al., 2022; ), or (iii) derived indirectly from the triggering events (Antonini et al., 2002; Chau et al., 2003; Grant et al., 2018; Huang and Li, 2009; Kobayashi et al., 1990).

In the field, a qualitative assessment of areas with frequent rockfall occurrence may be achieved by detecting a high number of visible recent rockfall scarps and/or rockfall boulders and impact marks on ground, trees and objects. Broken trees, low vegetation or aisles in the forest cover may give a well indication for a high rockfall frequency. Depending on the type of lithology, the fresh look of rockfall scarps in the rock walls and rockfall boulders (Fig. 1A, B) may disappear within few years, making it difficult to estimate the timing of the events based on field data.

Quantitative data about the temporal occurrence of rockfalls can be gathered by a systematic review of various historical accounts and sources, including (i) chronicles and archives in churches, museums, police and fire departments, administrative and government institutions, (ii) newspapers, (iii) technical and scientific reports, (iv) internet alerts, (v) interviewing of eye-witnesses, (vi) examination of historic photographs, and (vii) monitoring of rock faces.

Rockfall descriptions in historical accounts may contain sufficient detail to determine the exact rockfall locations, including
release and deposit area. However, in other historical reports the location is only mentioned vaguely referring to segments of valleys/roads/trails/villages or regional toponyms. Often the informal (local) place names reported in the historical reports are not mentioned/published in official (topographic) maps. The collaboration with local habitants and/or police officers is essential in order to determine the exact location of the rockfalls (Guzzetti et al., 1994; Melzner, 2015; Melzner and Braunstingl, 2017; Melzner et al., 2020, Salvati et al., 2010). Historical rockfall records usually lack of a complete information
on both the timing of the event and the size of the rock, which are usually referred in relative terms (i.e., "very often", "large", "destructive/catastrophic event"), or are estimated in the field in pre-defined size classes. In case of rockfalls which are identifiable from the examination of photographs only, the time precision is reduced to the year or the decade (Stock et al., 2011; Wieczorek and Snyder, 2004, Wieczorek et al., 1992).



### 2.3 Rockfall statistics and applications

Frequency analysis of temporal patterns *of rockfalls* can be performed or displayed using *absolute numbers of rockfalls*, *cumulative number of rockfalls*, and *normalized cumulative number of rockfalls* per unit time. Heuristic and statistical analyses and tests are used to check and detect frequency changes in rockfall data samples and to investigate temporal trends and patterns in rockfall time series.

*The total number of rockfalls* per unit of time gives information about the rockfall rate, per year or per day. *Cumulative rockfall*
*time series* display cumulative numbers of rockfalls per year or per day and can be used to analyse trends in rockfall occurrence and detect changes in the collection procedures, or presence of discontinuities. *Normalized cumulative rockfall series* display the (cumulated) frequency of rockfall divided by the total number of rockfall**.** These plots are used to detect changes in rockfall numbers across multiple rockfall series. This is an important step in the evaluation of the completeness or representativeness of data of a historical rockfall catalogue.

Change point and structural breaks in rockfall time series can be detected heuristically or automatically using different statistical tests. The literature on this topic is vast but mostly referring to applications on continuous time series. The Pettitt's test (Pettitt, 1979) is largely applied for detecting change points. In this test, the H0 (null hypothesis) corresponds to the absence of change points. If the p-value is less than 0.05, the null hypothesis is rejected which means that there is a change in rockfall numbers. This corresponds to the k-value as shown by the Pettitt plot**.** If the p-value is greater than 0.05, the null hypothesis is
accepted which refers that there are no significant changes in rockfall numbers. The detection of structural changes can be conducted using the Rec-CUSUM (Cumulative SUM of Recursive Residuals) method (Brown et al., 1975). Possible variations of rockfalls over time can be detected by significant monotonic upward or downward trends over time. The Mann-Kendall (Gilbert, 1987; Kendall, 1975; Mann, 1945) test is used as an alternative to a parametric linear regression analysis, which tests if the slope of the estimated linear regression line fitting the timeseries data is statistically different from zero.

The magnitude of a rockfall time series trend, i.e. quantifying the rate of increase or decrease of rockfalls over time, can be evaluated with a simple non-parametric procedure called Sen' slope (Sen, 1968). The test is based on the calculation of the Sen' slope statistics expressing the linear rate of rockfall change. Sen' slope may be lower and greater than zero indicating respectively the rate of the decreasing or increasing trend.

Temporal patterns analysis and trends and change points detection are important steps in the evaluation of the completeness
or representativeness of data of a historical rockfall catalogue, including the detection of instrumental/measuring errors or significant process changes.





## 3 Methodology

### 3.1 Collection of temporal rockfall data

We used seven catalogues of historical rockfalls in Austria, Italy and the USA to analyse the impact of mapping method and/or
strategy. Some of the catalogues are subsets of comprehensive rockfall inventories including detailed information about other
relevant rockfall information, such as volumes and structural geologic factors (Table 1). Different data collection strategies
were adopted to collect these temporal rockfall data.

The *inventory $I_H$* covers an area of about 17 km² in Austria, which is built up by carbonatic sedimentary rocks. The inventory
contains besides the *historical rockfall catalogues $C_{H8}$, $C_{H9}$ and $C_{H10}$* a variety of information (see catalogues on rockfall size
$C_{1-7}$ in Melzner et al., 2020). The *historical catalogue of rockfalls $C_{H8}$* (compiled in 2014) contains information about 76
historical rockfalls for the 362-year period from 1652 until 2014. This detailed catalogue was prepared by a systematic analysis
of different historical archives of the community, museum, police department and collaboration/interviews with eye-witnesses
respectively. Quantitative information about volume for most of these historical rockfalls is not existent. The qualitative
information about volume and damage were interpreted in terms of rockfall intensity and stored in the *catalogue of rockfalls
consequences $C_{H10}$* (Melzner, 2017, 2015; Urstöger, 2000). The *historical catalogue of rockfalls $C_{H9}$* (compiled in 2016)
captures 28 historical rockfalls within a progressive rockfall scarp alongside a deep-seated lateral rock spread for the three-
year period December 2013 until 2016. The catalogue was compiled by the analysis of information from a diary of a local
habitant (hunter) who lives permanently in the vicinity of that slope failure (which is located in a remote area). The eye-witness
only documented in his diary large rockfalls by listening loud noise, absolute volume information is therefore not existent
(Melzner, 2016).

The *inventory $I_{YV}$* covers an area of about 3000 km² in the USA, which is built up by igneous rocks. The inventory contains an
extensive catalogue of rockfall sizes (see catalogue of rockfall size $C_{YV}$ in Melzner et al., 2020) and the *historical catalogue
of rockfalls $C_{YV1}$*. This catalogue contains information about 887 historical rockfalls for the 154 years period from 1857 until
2011. The assessment strategy focused on the review of published and unpublished historical accounts, scientific reports and
papers, eyewitness observations, field mapping and remote sensing analysis (Wieczorek and Snyder 2004; Stock et al. 2011).

The country-wide *inventory $I_{AVI}$* covers the area of Italy of about 300.000 km², which is built up by large variety of rock types.
The *historical catalogue $C_{AVI}$* covers a time period from 1489 to 2001 with a total number of entries of about 2612 rockfalls.
The technique used to compile the historical rockfall catalogue AVI for the whole of Italy was towards rockfalls that resulted
in damage, affected buildings and roads and caused fatalities (Guzzetti & Tonelli, 2004).

The *inventory $I_{SAL}$* covers an area of about 1300 km² in Austria. The study area was chosen to contain a wide- range of different
topographic and geologic settings to cover a large variety of different processes. The inventory was compiled by the systematic
review of analogue police chronicles (Melzner, 2016) and the researched information could be classified in eleven process
types: fluviatile processes (FP), gravitational mass movements (GM), snow processes (SP), meteorological processes (MP),



seismic processes (S), and rockfalls (RF). The catalogue contains information about 690 historical rockfalls for the 162 years
period from 1854 until 2016.

## 3.2 Statistical analysis

We analysed the historical rockfall catalogues in terms of *numbers of rockfalls per year*, *normalized cumulative number of rockfalls per year* and *cumulative number of rockfalls per year*.

We calculated the *absolute numbers of rockfalls per year* for the catalogues $C_{AVI}$, $C_{YV}$, $C_{H8}$, $C_{SAL}$ and $C_{H9}$ to gain an overview
about the sequence of rockfalls which are reported in the different study areas in the time period 1857 to 2011. The annual or daily rockfall frequency $Nj$, can be determined by counting the number of rockfalls per year, or per day. Furthermore, we analysed the relationship of the documentation of rockfall occurrence (historical rockfall catalogues $C_{H8}$) with damage (catalogue of rockfalls consequences $C_{H10}$) in the time period 1857 to 2011.

We calculated the *cumulative number of rockfalls per year* for the catalogues $C_{AVI}$, $C_{YV}$, $C_{H8}$, $C_{SAL}$ and $C_{H9}$ to analyse
differences in datasets, e.g., rate of rockfalls per year for the time period 1489 to 2011 and for the time period 1857 to 2011. The statistical analysis is conducted by (i) counting the rockfalls per unit of time (e.g., a month, a year) to obtain a continuous time series, (ii) cumulating this new time series, and then (iii) analysing the rates of increase/decrease of the slope of the cumulative series over time (i.e., evaluating the first derivative). The angular coefficient (slope) of the linear fitting of a period gives the average number of rockfalls mapped every year in this period. Such coefficient basically quantifies the rate at which
rockfalls were reported in different portions of the time series.

We analysed *the normalized cumulative number of rockfall* for the catalogues $C_{AVI}$, $C_{YV}$, $C_{H8}$, $C_{SAL}$ and $C_{H9}$ and of the related slopes for the time period 1489 to 2011 and for the time period 1857 to 2011. The statistical analysis was conducted by calculating the cumulative number of rockfalls and dividing it by the total number of rockfalls (frequency distribution of rockfall scaled from 0 to 1). The analysis of the normalized cumulative number of rockfall and of the related slopes shows
more clearly the difference in statistics *among* the datasets.

We delineated *different periods of rockfall occurrence* using a *heuristic approach* and *statistical test*. We applied the Pettitt test to verify the presence of changes in the yearly rockfall time series. The test, compared to heuristic analyses, allows for a more objective and robust detection of the changes of the rockfall datasets. Specifica1ly, we used the Pettitt' test to detect the statistically relevant changes in the time series average numbers. This was conducted on the yearly rockfall counts series and
using a specific procedure coded in R language using the *package trend*.

We analysed the *relationship between daily and seasonal variability of high magnitude rockfalls* occurring alongside a deep-seated active lateral rock spread with respect to climatic time series. The rockfall time series is a subset of catalogue $C_{H9}$ for which meteorological data of a nearby weather station (source: eHYD - BMLRT and BEV, 2021) was available for the period December 2014 to December 2015 (Melzner et al., 2017).



Lastly, we analysed the *relationship between cumulative number of different processes and rockfalls which resulted in consequences* by calculating cumulative number of processes per year and mosaic plot. Such a plot enables the visualization of data sets with two or more variables. It provides an overview of the data and makes it possible to identify correlations or co-occurrences between the different characteristics. In the present example we displayed the Catalogue $C_{SAL}$ and show the relationship of consequences and rockfalls and other processes are analysed using numbers of rockfalls causing fatalities.

## 4 Results


The results of the comparison of the numbers of rockfalls per year ($D_R$) in all datasets indicate a general increase of reported rockfalls (Fig. 2). The country-wide catalogue $C_{AVI}$ contains the highest number of reported rockfalls. Catalogues $C_{YV}$ and $C_{H8}$ cover comparable area sizes, but the $C_{YV}$ shows a much higher number of reported rockfalls. Catalogues $C_{H9}$ contains the lowest number of reported rockfalls mainly concentrated in two-time periods.

The comparison of the historical rockfall catalogue $C_{H8}$ and the rockfall catalogue of damage $C_{H10}$ (Table 1 and Fig. 2) reveals that very old rockfalls before 1920 were only documented, if houses were destroyed. Most of the recorded rockfalls after 1920 caused severe damage (red and orange points) or resulted in fatalities (blue points) and injured persons (pink point). Since 2000, the rockfalls have been well documented which did not cause severe damage.

The comparison of the rockfall time series $C_{H8}, C_{H9}, C_{YV}, C_{AVI}$ and $C_{SAL}$ (Fig. 4) reveals differences among the datasets, which

indicates that the data collection method/source of information affects slopes of rockfall rates of the temporal rockfall datasets. The periods of rockfall rates detected for the different datasets do not correspond with each other: for $C_{YV}$, $C_{H8}$, $C_{SAL}$ five periods, $C_{AVI}$ six periods and $C_{H9}$ three periods could be delineated (Table 2). The comparison of adjusted R², with the majority of values greater than 0.9, suggests that the heuristic procedure was able to capture correctly the linear trends in different periods of the data series. The change points detected using the Pettitt test (Fig. 5 and Table 3) show a general trend for all

catalogues towards the detection of less periods than detected with the heuristic methods: for catalogue $C_{H8}$ two change points, for $C_{YV}$ four change points, for $C_{SAL}$ one change point, for $C_{AVI}$ three change points and for $C_{H9}$ no change points could be identified (Table 3). The analyses reveals some correspondences with those obtained by the heuristic analysis. For $C_{AVI}$ the Pettitt' test detected three change points for the years 1877, 1950 and 1983, which may correspond with the heuristically detected breakpoints in the years 1890, 1950 and 1981 (Table 2). In case of $C_{YV}$, the statistical test detected three change points

in the years 1911, 1979 and 1996, that may correspond to the heuristically detected breakpoints in the years 1917, 1978 and 1995. For $C_{H8}, C_{H9}$ and $C_{SAL}$ the comparison does not highlight any correspondence (Table 3).

The analysis of the daily occurrence of high magnitude rockfalls at the deep-seated lateral spread (Fig. 6, 7) reveals that a total of 14 days during a one-year period were subject to rockfalls; of which two days with several rockfalls. The first rockfall occurred on December 27, 2013, followed by three or four-to five-day periods (12 to 15 February, 24 to 27 February, and 1 to

4 March, 2014) with a rockfall reported each day, several rockfalls occurred on February 15, 2014 (during the day) and March





4, 2014 (in the night), and another rockfall on March 27, 2014. In April and May 2014 on two-day periods (11.4.-12.4.2014 and 7.5.-8.5.2014) and on May 2$^{nd}$ 2014 each day a rockfall occurred. In 2015, the first rockfall occurred March 17$^{th}$, followed by two three-day periods (25.8.-27.8.2015 and 8.9.-10.9.2015). In 2016, two rockfalls were recorded on two single days (6.2.2016 and 31.8.2016) (Melzner, 2015). The comparison with the meteorological data does not reveal any significant
correlation.

The results of the analysis of catalogue C$_{SAL}$ (Fig. 8, 9) show that the majority of the events resulted in property damage, but 71 process events also claimed fatalities or injured persons (number: 100). For 62 event entries no information on property damage or personal injury was included, for 17 events it was explicitly noted that no personal injury or property damage was caused by the event. The personal injuries investigated were primarily caused by fluviatile processes. In the case of
gravitational mass movements, only rockfall resulted in fatalities: a total of 15 deaths and 10 injuries were recorded due to rockfalls. A total of 31 people was killed and 13 injured in snow avalanches. 14 dead's, 4 injured and more due to storm and thunderstorm (exact number of injured is not mentioned in two chronicle entries). Figure 9 displayed that major property damage is caused by floods, but that most deaths are caused by rockfalls.

## 5 Discussion

Following facts can be summarized for catalogue C$_{YV}$ (Fig. 4 and Table 2) concerning the completeness and representativeness of data: (i) no data was available before 1851, which is the discovery year of the Yosemite Valley, (ii) first data entries comprise only very large rockfalls, which were reported randomly by eye-witnesses, (iii) more systematical recording of small and large rockfalls affecting infrastructures began after ~1911-1916 to maintain damages, (iv) increasing tourist numbers and thus higher risk required are more sytematical collection of rockfall data resulting in a higher number of reported rockfalls, (v) a rockfall
event killing a few people in the early 1980s resulted in the establishment of a rockfall data collection programm using archives, aerial photo interpretation and field work, and (vi) the high rockfall rates in recent years are due to very detailed documentation of an investigator permanently living in that area (Stock et al., 2011; Wieczorek and Snyder, 2004).

In contrast, the analysis of the completeness and representativeness of the catalogue C$_{H8}$ time series (Fig. 4 and Table 2) showed that (i) in the first period only rockfalls were reported, which damaged houses or infrastructures or had large
magnitudes, (ii) in the second period nearly every documented rockfall resulted in damages (Fig. 4), (iii) in the last, most recent period rockfalls are included which did not cause damage, (iv) type and extent of damage were most of the times provided, volume was reported in a qualitative manner, (v) significant changes in the abundance and distribution of the elements at risks, including the population, are important factors influencing the rockfall rates, and (vi) the effect of preventive measurement are only in a very few parts in the last period relevant (Melzner, 2017, 2015; Melzner et al., 2019). In contrast to C$_{YV,}$ the area has
a very long settlement history, thus very old rockfalls could be recorded by eye witnesses.



The analysis of the completeness and representativeness of the catalogue $C_{AVI}$ (Fig. 4 and Table 2) showed that with a systematic collection of data focusing on rockfalls which resulted in consequences, a very comprehensive countrywide database could be established. The following facts can be stressed: (i) journals emphasize large magnitude rockfalls, (ii) information about rockfalls improved after World War II, and markedly increased during the 1950s (i.e. change detected by

both heuristic and statistical test), after journals introduced regional and local chronicles, (iii) newsletter articles provided quality data on the dates but not the time and the triggering mechanism of occurrence, (iv) the exact location was rarely reported and only for single, large rockfalls, (v) economic estimates of the type and extent of damage were provided in a few articles; material and volume were seldom reported, (vi) review of technical and scientific documents provide high-quality data for a small number of events, rarely describe extent of damage, or contain social and economic consideration, and (vii) interviews

with experts provide general information on a limited number of rockfalls (Guzzetti et al., 1994; Salvati et al., 2010).

The European datasets ($C_{AVI}$, $C_{SAL}$ and $C_{H8}$) show a common increase between 1950 and 1956 that could be possibly related to the increase of data sources availability in the period of recovery and reconstruction after the Second World War, which did not affect the USA ($C_{YV}$). This change was detected by both heuristic and statistical procedures. In all the datasets, except the last period of rockfall rates of $C_{SAL}$, the rockfall frequency increased with some fluctuations. Catalogue $C_{H9}$ forms an exception

being a progressive rockfall failure, showing two periods of higher rockfall rates.

The main findings on the comparison of the catalogues of historical rockfalls can be summarized as follows: constant, high rates of yearly cumulative number of rockfall rate are recognizable, if the data was collected continuously by an expert, or the data was collected with the internet and social media. Low rates of cumulated rockfalls can be justified by (i) large magnitude rockfalls being preferentially reported, number of smaller rockfalls underestimated, (ii) rockfalls with no consequences

(fatalities and damage) are underrepresented, (iii) rockfalls in remote areas, and (iv) rockfalls during night are often not recorded or reported. Recurring visits of rockfall areas and a close contact with local people/authorities raises the number of reported rockfalls. Geographic location of information is very accurate, if the rockfall resulted in a consequence. In very old historical sources, the geographical location of a rockfall is given by specific local names which do not appear on recent maps. Further, in the Alps, it is common to name places or single houses after the name of residents, which are not named in base

data such as topographic maps. An accurate geographical positioning of this information is only possible with the support by the knowledge of local habitants or experts. A personal relation to the local people due to recurrence visits raise the results of data acquisition. Common to historical rockfall records is that estimates of rockfall size is incomplete, or that size estimates are given in relative terms (i.e., small, large, very large), or are estimated in pre-defined size classes. This makes it very difficult to establish frequency-size relations the data being biases.

The analysis of the relationship of the catalogue $C_{H9}$ with the meteorological data does not show any significant correlation. This might be due to the fact, that the weather station is located a few hundred meters lower down in the valley. Or this might indicate that rockfalls were initiated by the acceleration in the lateral spreading of the deep-seated slope deformation (Melzner et al. 2017), thus giving relative information about the activity phases of this deep-seated rock spread. High frequency low





magnitude rockfalls occur often in this active spreading area and were not recorded systematically. The analysis of an optic

photo monitoring system showed that this method is more suitable to detect large volume rockfalls. Especially the fast-changing climatic condition (light, fog, rain) in such a high alpine area makes it difficult to apply semi-automatic or automatic analysis methods for the detection of rockfalls.

Finally, catalogue C$_{SAL}$ showed that a systematic review of police chronicles for a selected pilot area (covering different geological units and reliefs) allows to establish a very comprehensive database of different processes (Melzner et al. 2012,

Melzner, 2016; Melzner and Braunstingl, 2017), although non-damaging rockfalls are still underrepresented. Chronicles of police departments in Austria provide squeezed information about the susceptibility of a (large) region towards different natural processes and risk data. The subjective feelings and interests of the police inspectors have a strong effect on the level of detail of the entries. Most of the natural events in this catalogue resulted in damage and personal injury.

## 6 Conclusions

The estimation of the temporal frequency of the involved rockfall processes is an important part in hazard and risk assessments. Different methods can be used to collect and analyse rockfall data. Depending on lithology, recent rockfall scarps or rockfall boulders may rapidly lose their fresh appearance or are removed by the habitants, which makes it difficult to estimate the time of occurrence of rockfalls and a corresponding return period of the rockfalls in the field. Monitoring of rockwalls in Alpine relief is most of the times not practicable and affordable by the communities.

From a data collection point of view, research activities in different archives enables the compilation of rockfall inventories and rockfall catalogues (with information on data source and definition of quality criteria). The smart use of the data (e.g., a careful selection of representative samples) is of high importance in hazard and risk assessments.

From a statistical point of view, rockfall datasets are nearly always incomplete. Simple heuristic and statistical frequency-based methods can be used for the basic rockfall catalogue analysis and characterization, however additional non-parametric

methods have to be developed to cope with less data entries and close data gaps and to support the identification of a shared standard methods for rock fall time series analysis. The detection and handling of breakpoints/change points may avoid inappropriate use of data series and related models in hazard and risk analysis.

Accurate data collection approaches and the application of statistical methods on existing rockfall data series as reported in this study should be better considered in rockfall hazard and risk assessments in the future.

7 Acknowledgements

The Geological Survey of Austria is thanked for supporting S.M. during the paper completion also after leaving the Geological Survey. Special thanks Dr. Fausto Guzzetti, Paola Reichenbach, Mauro Cardinali, and Paola Salvati (CNRI, IRPI, Perugia) for



several discussions on that topic and the AVI dataset. G. F. Wieczorek (USGS) and G. Stock (NPS) for the discussions, the invitation to the Yosemite National Park, and the Yosemite rockfall dataset. The data acquisition was conducted in the frame

of the GEORIOS project by the Geological Survey of Austria (GBA), NARIS project (funded by GBA and the Federal State Government of Salzburg). Climatic data was provided by eHYD. Special thanks go to the community of Hallstatt (in particular K. Wirobal and H. Urstöger from the Museum Hallstatt, Stefan Janu, Friedrich Idam and Klaus Reisenauer) for providing information about rockfall events and the Salzburg State Police Department and the employees of the various police inspections supporting the analysis of the police chronicles. Thank you to CNR-IRPI for funding the open access publishing of the present

article.

## 8 Authors contribution statements

SM: conceptualization and coordination, data collection, selection of data sets, statistical analysis of data, table/ figure drafting and preparation of figures, manuscript writing. MR: preparation of R codes and statistical analysis of data, table/ figure drafting, manuscript writing. MC: reviewing of final manuscript. JH: reviewing of final manuscript.

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



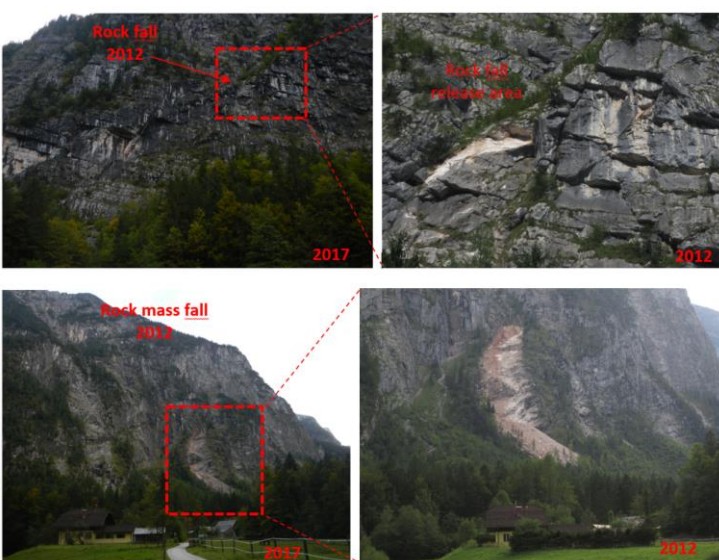

**Figure 1: Estimation of temporal occurrence/age of a rockfall or rock mass fall based on the rock colour in Dachstein limestone. In case of the rockfall (upper pictures) the rockfall release area and rockfall boulders have already after 3-4 years the same colour as the surrounding rock. In case of the large rock mass fall (lower pictures) similar colour transition is visible (photos by S. Melzner, GBA archive).**

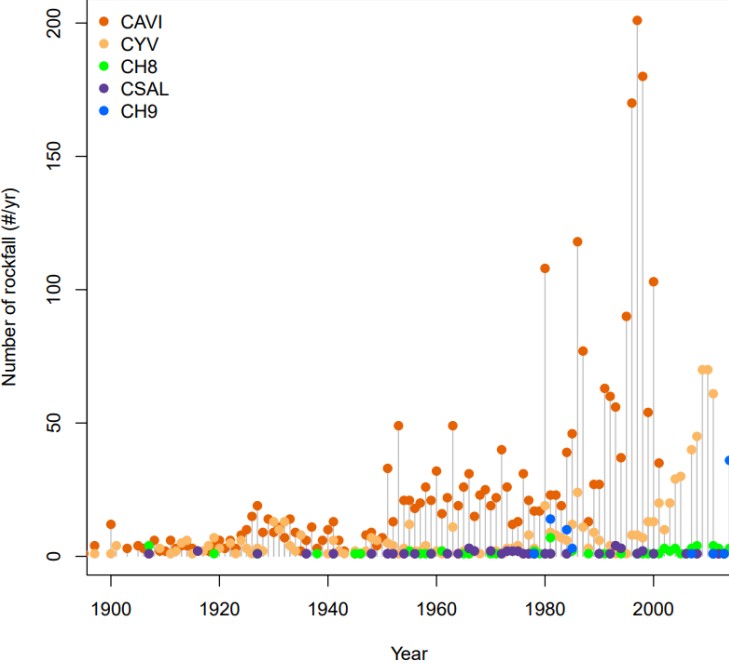

**Figure 2: Comparison of absolute number of rockfalls per year ($N_j$, (annual rockfall frequency) of five historical rockfall time series in Italy (red), Austria (green, purple and blue) and USA (orange). C$_{YV}$ catalogue 1857-2011 (total number of rockfalls 887), C$_{AVI}$ 1489-2001 (total number of rockfalls 2612), C$_{H8}$ 1652- 2014 (total number of rockfalls 76), C$_{SAL}$ 1907-2016 (total number of rockfalls 53) and C$_{H9}$ 1978-2016 (total number of rockfalls 41).**


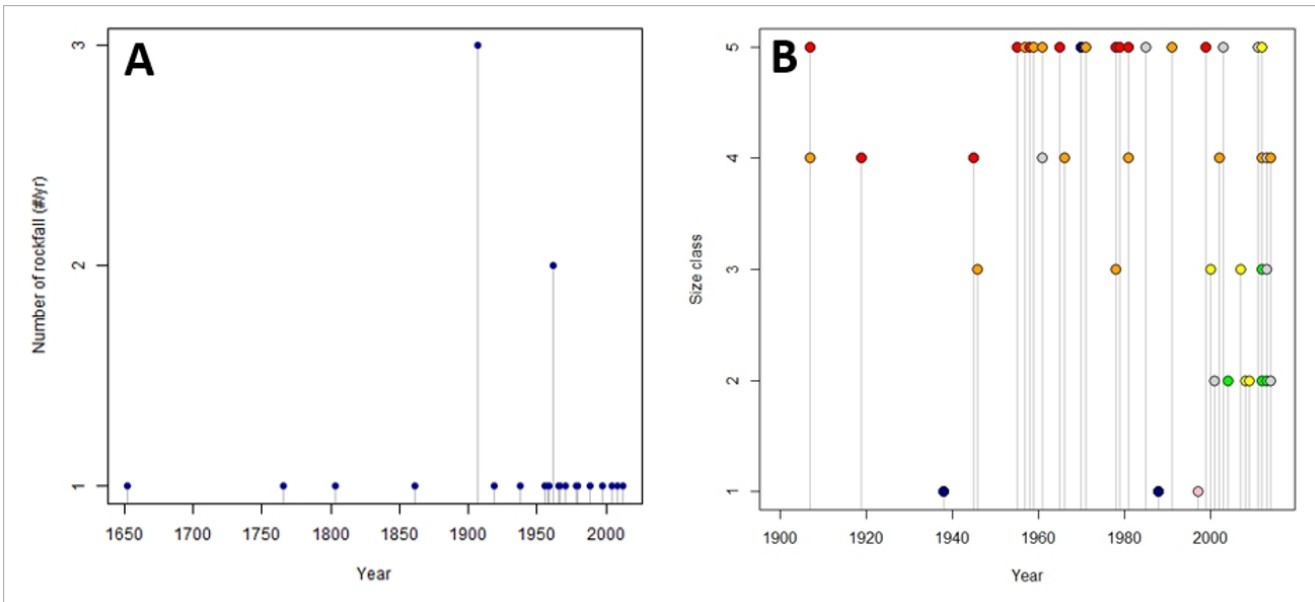

**Figure 3: Relationship between the representativeness of the rockfall time series $C_{H8}$ (A) with respect to rockfall which resulted in**
**consequences $C_{H10}$ (B). Legend of fig. B: red points=very large intensity, orange points=large intensity, yellow points=medium intensity, green points= low intensity, blue points= fatality, rosa points= injury, grey points= no info, grey points= no damage.**






**Figure 4: Comparison of the cumulative number of rockfalls (A & B) and normalized cumulative number of rockfalls** (N_CR, as a function of year, C & D) of the five historical rockfall catalogues $C_{H8}$, $C_{H9}$, $C_{YV}$, $C_{AVI}$ and $C_{SAL}$ from USA (orange curve), Italy (red curve) and Austria (dark and light blue and green curve).

**Figure 5: Results of the Pettitt test for the different rockfall datasets, A= C$_{H8}$, B= C$_{YV}$, C= C$_{AVI}$, D= C$_{SAL}$. Vertical dashed blue lines show the change points detected by the test. Datasets without detected change points are not shown.**





**Figure 6: High magnitude rockfalls occurring always in the same rockfall source area (progressive failure) alongside a large rock spread (Photos taken 07/2013 and 09/2021 by S. Melzner, see Melzner et al. 2017).**




**Figure 7: Comparison of rockfalls with climatic conditions (new snow, snow heights, rainfall, temperature). Rockfall data collected by questionnaire/diary of an eye whiteness (a hunter) who is permanently living in this remote area. Data represents only high magnitude rockfalls always in the same rockfall source area (progressive failure).**


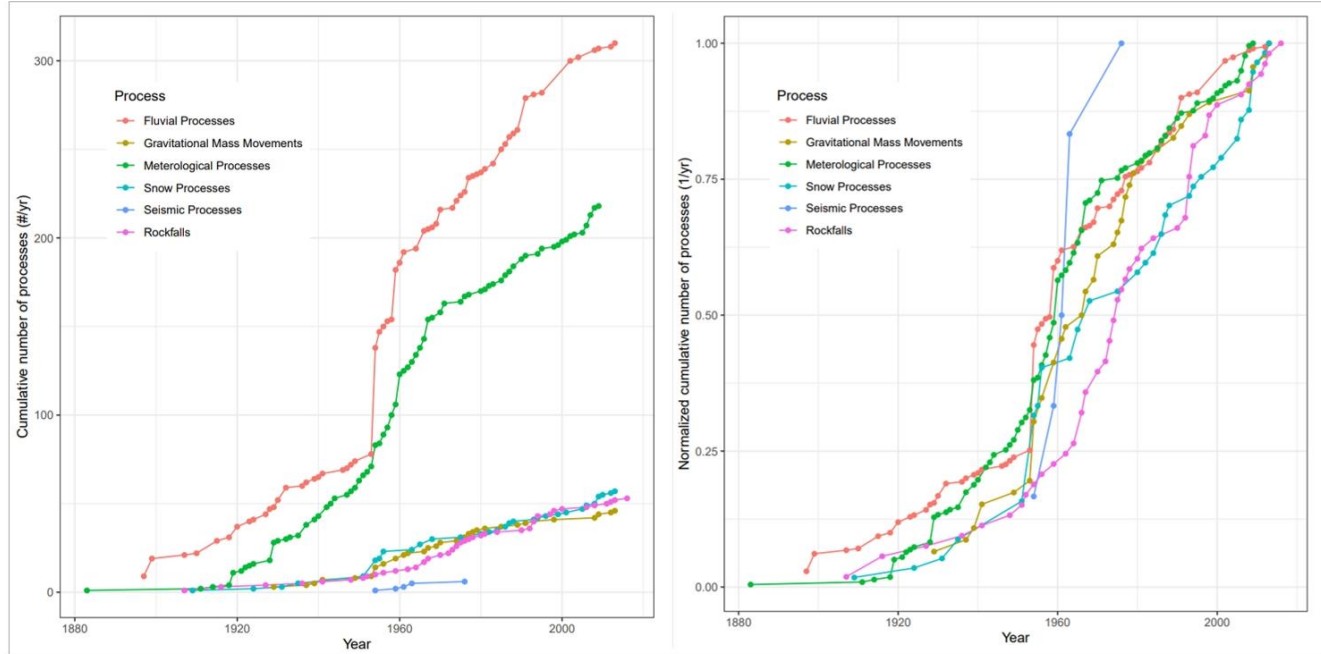

**Figure 8: Comparison of the cumulative number of different natural processes ($N_{CR}$, as a function of year) of the catalogue compiled by the review of the chronicles of the police department in Austria.**






**Figure 9: Display of relationship consequences and natural processes of the catalogue compiled by the review of the chronicles of the police department in Austria. P/D= damage and fatalities, D= damage, P= fatalities, N/A=no information in the historical account, N/D= no consequences. FP= Fluvial processes (light blue), GM=Gravitational Mass Movement (brown), MP= Meteorologic Processes (blue), SP= Snow Processes (grey), S= Seismic Processes (red) and RF= Rockfall (green).**






**Table 1: Overview about catalogues of historical rockfalls used for statistical analysis.**

| Inventory | Catalogue | Country | Area size [km²] | Number of data entries | Time period | Rock type | Mapping strategy/data collection method | Source of rockfall information | Reference |
|---|---|---|---|---|---|---|---|---|---|
| $I_H$ | $C_{H8}$ | AUT | 6,7 | 76 | 1652-2014 | Carbonatic, sedimentary rocks | Community chronicle, police department, questionnaire's habitants | Archive of the community, police department, private archives | (Melzner, 2015; Urstöger, 2000) |
| | $C_{H9}$ | AUT | 1 | 41 | 1976-2016 | Carbonatic, sedimentary rocks | Diary of local person | private diary | (Melzner, 2015; Melzner et al., 2017) |
| | $C_{H10}$ | AUT | 6,7 | 76 | 1652-2014 | Carbonatic, sedimentary rocks | Community chronicle, police department, questionnaire's habitants | Archive of the community, police department, private archives | (Melzner, 2015; Urstöger, 2000) |
| $I_{YV}$ | $C_{YV}$ | US | 3000 | 317 | 2011 | Igneous rocks | Published and unpublished historical accounts and field studies | | (Stock et al., 2011; Wieczorek et al., 1992; Wieczorek and Snyder, 2004) |
| | $C_{YV1}$ | US | 3000 | 887 | 1857-2011 | Igneous rocks | Local National Park rangers, USGS geologists | Reports of National Park, publications, private archives | Stock et al., 2011; Wieczorek et al., 1992; Wieczorek and Snyder, 2004 |
| $I_{AVI}$ | $C_{AVI}$ | IT | 300.000 | 2612 | 1489-2001 | Various rocks | Old newspapers, chronicles | | (Guzzetti et al., 1994; Salvati et al., 2010) |
| $I_{SAL}$ | $C_{SAL}$ | AUT | 1300 | 690 | 1849-2014 | Various rocks | Systematic analysis of all analogue chronicle of the police departments | chronicle of the police departments | (Melzner, 2016; Melzner and Braunstingl, 2017) |








**Table 2**: **Results of the heuristic manual analysis of the cumulative rockfall series. For the different catalogues (C$_{H8}$, C$_{YV}$, C$_{AVI}$, C$_{SAL}$, C$_{H9}$) the heuristic analysis of the slope of the cumulative rockfall frequency enabled the delineation of different periods. For each period the slope ("Estimated rockfall rate") gives information on the yearly average number of mapped rockfalls. Adjusted R$^2$ quantifies the degree of fitting of the linear model assumed for fitting the data in the different periods.**

| Catalogue | Period | Start period | End period | Estimated rockfall rate | Standard Error | Adjusted R$^2$ | Description of rockfall information in the different rockfall catalogues |
|---|---|---|---|---|---|---|---|
| C$_{YV}$ | 1 | 1857 | 1917 | 0.642 | 0.048 | 0.907 | First data entries comprise only very large rockfalls, which were reported randomly by eye witnesses |
| C$_{YV}$ | 2 | 1918 | 1938 | 4.627 | 0.291 | 0.944 | More systematically recording of small and large rockfalls affecting infrastructures began after 1911-1916 to maintain damages |
| C$_{YV}$ | 3 | 1939 | 1978 | 2.778 | 0.066 | 0.982 | |
| C$_{YV}$ | 4 | 1979 | 1995 | 7.719 | 0.685 | 0.906 | A rockfall event killing a few people in the early 80s resulted in the establishment of a rockfall data collection program using archives, aerial photo interpretation and field work |
| C$_{YV}$ | 5 | 1996 | 2001 | 11.914 | 1.130 | 0.957 | Detailed documentation of an investigator permanently living in that area |
| C$_{YV}$ | 6 | 2002 | 2011 | 39.064 | 3.561 | 0.937 | Detailed documentation of an investigator permanently living in that area |
| C$_{H8}$ | 1 | 1652 | 1890 | 0.027 | 0.006 | 0.776 | Only rockfalls that damaged houses, infrastructures or had large magnitudes |
| C$_{H8}$ | 2 | 1891 | 1955 | 0.153 | 0.013 | 0.954 | Nearly every rockfall resulted in damages |
| C$_{H8}$ | 3 | 1956 | 1980 | 0.458 | 0.030 | 0.963 | |
| C$_{H8}$ | 4 | 1981 | 2000 | 0.429 | 0.029 | 0.965 | |
| C$_{H8}$ | 5 | 2001 | 2014 | 2.080 | 0.056 | 0.992 | Information about rockfalls are as well included which didn't cause damages |
| C$_{AVI}$ | 1 | 1489 | 1890 | 0.112 | 0.006 | 0.917 | |
| C$_{AVI}$ | 2 | 1891 | 1920 | 2.633 | 0.093 | 0.978 | |
| C$_{AVI}$ | 3 | 1921 | 1942 | 9.486 | 0.258 | 0.985 | |
| C$_{AVI}$ | 4 | 1943 | 1950 | 4.680 | 0.755 | 0.862 | Information improved after World War II |
| C$_{AVI}$ | 5 | 1951 | 1980 | 24.721 | 0.312 | 0.995 | Information markedly increased during 1950s, after journals introduced regional and local chronicles |
| C$_{AVI}$ | 6 | 1981 | 1988 | 53.512 | 5.659 | 0.927 | |
| C$_{AVI}$ | 7 | 1989 | 2001 | 100.967 | 6.613 | 0.951 | Internet alerts were included into the documentation of rockfalls |
| C$_{SAL}$ | 1 | 1907 | 1963 | 0.204 | 0.019 | 0.909 | Only rockfalls resulting in damage or fatalities are recorded in the chronicle |
| C$_{SAL}$ | 2 | 1964 | 1980 | 1.154 | 0.052 | 0.978 | Only rockfalls resulting in damage or fatalities are recorded in the chronicle |
| C$_{SAL}$ | 3 | 1981 | 1990 | 0.214 | 0.041 | 0.929 | Only rockfalls resulting in damage or fatalities are recorded in the chronicle |
| C$_{SAL}$ | 4 | 1991 | 1997 | 1.429 | 0.603 | 0.606 | Only rockfalls resulting in damage or fatalities are recorded in the chronicle |
| C$_{SAL}$ | 5 | 1998 | 2016 | 0.376 | 0.036 | 0.940 | Only rockfalls resulting in damage or fatalities are recorded in the chronicle |
| C$_{H9}$ | 1 | 1978 | 1984 | 4.0 | 0.385 | 0.982 | Progressive rockfall failures alongside a deep-seated lateral rock spread |
| C$_{H9}$ | 2 | 1985 | 2012 | 0.067 | 0.027 | 0.724 | |
| C$_{H9}$ | 3 | 2013 | 2016 | 14,2 | 5.637 | 0.640 | High magnitude rockfalls in a progressive rockfall failure alongside a deep-seated lateral rock spread |





**Table 3**: **Results of the Pettitt test for the detection of change points in rockfall frequencies in temporal rockfall catalogues.**

| Catalogue | Period | Start period | End period | Pettitt' test statistics (U-value) | Pettitt' test significance (p-value) | Change point (year_change) |
|---|---|---|---|---|---|---|
| $C_{H8}$ | 1 | 1652 | 2014 | 9326 | < 0.05 | 1937 |
| $C_{H8}$ | 2 | 1937 | 2014 | 710 | < 0.05 | 1996 |
| $C_{YV}$ | 1 | 1857 | 2011 | 3873 | < 0.05 | 1911 |
| $C_{YV}$ | 2 | 1911 | 2011 | 1399 | < 0.05 | 1979 |
| $C_{YV}$ | 3 | 1979 | 2011 | 203 | < 0.05 | 1998 |
| $C_{YV}$ | 4 | 1998 | 2011 | 45 | < 0.05 | 2006 |
| $C_{AVI}$ | 1 | 1489 | 2001 | 40010 | < 0.05 | 1877 |
| $C_{AVI}$ | 2 | 1877 | 2001 | 3722 | < 0.05 | 1950 |
| $C_{AVI}$ | 3 | 1950 | 2001 | 474 | < 0.05 | 1983 |
| $C_{SAL}$ | 1 | 1907 | 2016 | 1137 | < 0.05 | 1950 |
| $C_{H9}$ | 1 | 1978 | 2016 | 137 | ≥ 0.05 | No significant change points detected |
