# Peer review of "Lessons learnt from rockfall time series analysis: data collection, statistical analysis and applications"

_Natural Hazards and Earth System Sciences, 2023_

## Referee Comment (RC2)

General comments

- **Introduction.** The introduction lacks an overview of rockfall inventories and/or catalogues, as well as examples showcasing their importance for hazard assessment studies.
- **Methodology**. In the definitions section, the authors discuss the concepts of completeness and representativeness of the collected data. Have rockfalls that occurred but were not included in the catalogs been checked, evaluated or considered in any way? This aspect is particularly critical when considering catalog records gathered by non-experts, such as citizens, police officers, or hunter.
- **Methodology**. Why was a time interval from 1857 to 2011 selected for calculating the absolute numbers of rockfalls per year in the CAVI, CYV, CH8, CSAL, and CH9 catalogs, when we have earlier and later data available in the catalogs? Also, why was this time interval chosen for calculating the cumulative number of rockfalls per year? Please clarify it.
- **Methodology**. Please explain why the CYV1 catalog has been excluded from the statistical analysis. In many parts of the text, reference is made to the CYV catalog for a time series starting from 1851, when it only covers year 2011 (i.e. line 230 or foot of Figure 2). Please review this.
- **Results.** Regarding the analysis of the correlation between meteorological data and rockfalls, I don't understand why the authors have chosen to focus only on one catalog for this study (CH9), instead of using several catalogs as they have done in the analysis of change points and structural breaks in the time series. In my opinion, if the analysis is expanded to include other catalogs, it could be very interesting point to discuss. Otherwise, in my opinion, the inclusion of this part of the text seems inconsistent with the rest of the article.
- **Results.** It is understood that authors, in order to detect trend lines in time series, have used all the records from the catalogs for subsequent analysis. Therefore, the rate and change points (especially in catalogs with few records) can capture not only a higher frequency of rockfalls but also more systematic recording. To analyze the frequency, have the authors not considered using additional information, such as the size of the boulders, by comparing different scenarios (medium, large, or very large size)?
- **Discussion.** The catalogs used have very different time scales (ranging from only one year to 500 years) and spatial scales (ranging from 1 $km^2$ to 300,000 $km^2$). It would be interesting to include in the discussion section how these particularities have influenced the statistical study and how they may impact the overall conclusions.
- **Conclusion.** It would be interesting to have some general concluding remarks that should be taken into account in future rockfalls inventories/ catalogues to improve the results and lessons learned from this study.

Minor comments

- Line136: In Table 1, catalog CH9 displays 41 records. Please clarify why the text indicates 28 records.
- Line192: The text states, "Catalogues CYV and CH8 cover comparable area sizes, but the CYV shows a much higher number of reported rockfalls." However, it is worth noting that the CYV covers an area of 3000 $km^2$, while CH8 only covers 6.7 $km^2$. Please clarify.

- Line195: The reference Fig. 2 is incorrect; please change it to Fig. 3.
- Table 1: For a better understanding of Table 1, it would be interesting to include a column indicating the type of information provided by each catalog, such as historical data, consequences, etc.
- Figure 9: The numbers overlap with the P/D rectangles. Please relocate them to another position.

---

## Author Response (AR1)

**Review NHESS-2023-10**

**Editor**

Dear Editor,

We revised our manuscript according the comments of the two reviewers. You will find indications of the changes done in the second section of this document.

Furthermore, we adapted the formatting of references according the journal requirements. We used the color-blind tool to transform our figures. Due to the low quality of the figures we put the color blind figures in the supplementary materials.

Lastly, we corrected some spelling errors and English grammar.

Please let us know if you need further information.

Best regards

Thanks in advance

Sandra Melzner of behalf of all the co-authors

**Reviewer 1**

**General comments:**

Very interesting statistical analyses of different rockfall catalogues and the resulting conclusions and relationships.

Dear Reviewer. Thank you very much for your efforts reviewing our paper and for your valuable technical input.

**Specific comments:**

An interesting aspect would be to look at the data from the IFFI (Inventario dei fenomeni franosi d'Italia), available and downloadable from the IdroGEO platform, and compare it with the data sets of this study. The IdroGEO platform is for example up to date for the Autonomous Province of Bolzano and contains numerous, also very detailed records and data of rockfalls. Presumably, a change point in the 2000s could also be traced back to digitalisation and the simplified reporting of events (letter correspondence / fax vs. emails).

Thank you very much for this very interesting advice. It would certainly be very exciting to include the data. For the present work, however, we focused on datasets, for which we were involved in data collection and/or data analysis. We are aware that the data you mention is very interesting and could be used for future analysis and applications (we already did some analysis for the Federal State Government of South Tyrol).

When drawing conclusions, one could possibly take into account that experience shows that a whole series of events occur especially during major events (storm fronts with high intensities), whereby the smaller, at that time irrelevant events usually get lost in the background or are not considered due to the prevailing state of emergency and are thus not taken into account in the data collection. This is certainly also an aspect of the completeness / incompleteness of rockfall catalogues and data sets.

We will add this aspect to the conclusions.

**Technical Corrections:**

- Line 195: [...] The comparison of the historical rockfall catalogue CH8 and the rockfall catalogue of damage CH10 (Table 1 and Fig. 2) reveals [...] Fig. 3 not Fig. 2 or?

Yes, correct. We will refer to figure 3.

**Reviewer 2**

**General comments**

• Introduction. The introduction lacks an overview of rockfall inventories and/or catalogues, as well as examples showcasing their importance for hazard assessment studies.

Dear Reviewer. Thank you very much for your efforts reviewing our paper and for your valuable technical input.

This article is not a literature review, that's why some other examples are cited in the publication (we have reached already the maximum page limit of 24 pages). In any case we already cited reference containing a robust analysis of the literature (e.g., Melzner et al., 2020).

• Methodology. In the definitions section, the authors discuss the concepts of completeness and representativeness of the collected data. Have rockfalls that occurred but were not included in the catalogs been checked, evaluated or considered in any way? This aspect is particularly critical when considering catalog records gathered by non-experts, such as citizens, police officers, or hunter.

These datasets are considered to be representative. The police chronicle is supposed to be complete for damage-triggering events as recorded by the police officers.

• Methodology. Why was a time interval from 1857 to 2011 selected for calculating the absolute numbers of rockfalls per year in the CAVI, CYV, CH8, CSAL, and CH9 catalogs, when we have earlier and later data available in the catalogs? Also, why was this time interval chosen for calculating the cumulative number of rockfalls per year? Please clarify it.

The time interval was selected, because for this time spam data was available for all datasets, making a comparison possible.

• Methodology. Please explain why the CYV1 catalog has been excluded from the statistical analysis. In many parts of the text, reference is made to the CYV catalog for a time series starting from 1851, when it only covers year 2011 (i.e. line 230 or foot of Figure 2). Please review this.

You're right. This is a residual of a former version. We deleted this.

• Results. Regarding the analysis of the correlation between meteorological data and rockfalls, I don't understand why the authors have chosen to focus only on one catalog for this study (CH9), instead of using several catalogs as they have done in the analysis of change points and structural breaks in the time series. In my opinion,

if the analysis is expanded to include other catalogs, it could be very interesting point to discuss. Otherwise, in my opinion, the inclusion of this part of the text seems inconsistent with the rest of the article.

Good point. This paper gives examples for the potential use of rockfall datasets for statistical analysis. CH9 was chosen for the correlation with meteorological data because it is a representative dataset for such analysis, which none of the other datasets is.

We included now this point in the conclusions.

• Results. It is understood that authors, in order to detect trend lines in time series, have used all the records from the catalogs for subsequent analysis. Therefore, the rate and change points (especially in catalogs with few records) can capture not only a higher frequency of rockfalls but also more systematic recording. To analyze the frequency, have the authors not considered using additional information, such as the size of the boulders, by comparing different scenarios (medium, large, or very large size)?

Important topic to stress, we agree! Melzner et al. (2020) discusses the impact of mapping strategies on rockfall size distributions. Most records on historical rockfall data do not contain information on size. Often the information "on size" is given only in qualitative terms (i.e. "very big", " very destructive" etc.) and has to be interpreted by the expert (see example in Fig. 3). Most of the time "frequency-size distributions" are only possible on single cliffs, where a detailed monitoring system is installed.

We included this point in the conclusions

• Discussion. The catalogs used have very different time scales (ranging from only one year to 500 years) and spatial scales (ranging from 1 km2 to 300,000 km2 ). It would be interesting to include in the discussion section how these particularities have influenced the statistical study and how they may impact the overall conclusions.

Thanks for the advice, we included this in the conclusion section (see next comment).

• Conclusion. It would be interesting to have some general concluding remarks that should be taken into account in future rockfalls inventories/ catalogues to improve the results and lessons learned from this study. Minor comments

Thanks for this advice, we included some lessons learned in the conclusions.

• Line136: In Table 1, catalog CH9 displays 41 records. Please clarify why the text indicates 28 records.

Thanks a lot for the comment, 38 rockfalls occurred on 28 days, we changed this in the text.

• Line192: The text states, "Catalogues CYV and CH8 cover comparable area sizes, but the CYV shows a much higher number of reported rockfalls." However, it is worth noting that the CYV covers an area of 3000 km2 , while CH8 only covers 6.7 km2 . Please clarify.

Thanks for the advice, we changed the text to "Catalogues CYV and CH8 cover study areas with similar area settings (i.e. both glacially over-steepened trough valleys, one million tourists per year), but the CYV shows a much higher number of reported rockfalls."

• Line195: The reference Fig. 2 is incorrect; please change it to Fig. 3.

Thanks for the advice, we referred now to Fig. 3.

• Table 1: For a better understanding of Table 1, it would be interesting to include a column indicating the type of information provided by each catalog, such as historical data, consequences, etc.

Under "source of information" you find the required information.

• Figure 9: The numbers overlap with the P/D rectangles. Please relocate them to another position.

Thanks for the advice, we relocated them accordingly.